# Small Ruminant Production in Tanzania, Uganda, and Ethiopia: A Systematic Review of Constraints and Potential Solutions

**DOI:** 10.3390/vetsci8010005

**Published:** 2020-12-31

**Authors:** Bryony Armson, Abel B. Ekiri, Ruth Alafiatayo, Alasdair J. Cook

**Affiliations:** The Veterinary Health Innovation Engine (vHive), School of Veterinary Medicine, University of Surrey, Guildford GU2 7AL, UK; b.armson@surrey.ac.uk (B.A.); r.alafiatayo@surrey.ac.uk (R.A.); alasdair.j.cook@surrey.ac.uk (A.J.C.)

**Keywords:** Ethiopia, Tanzania, Uganda, small ruminants, constraints, solutions, goats, sheep

## Abstract

Sheep and goats are an important commodity for smallholder farmers across East Africa, but severe limitations remain in small ruminant production. This review aimed to identify specific constraints to small ruminant production and identify practical and sustainable solutions. From 54 eligible articles, most were focused in Ethiopia (n = 44) with only 6 studies performed in Tanzania and 4 in Uganda. The most frequently identified constraint in Ethiopia and Tanzania was disease (n = 28 and n = 3, respectively), and in Uganda, it was the lack of access to veterinary services (n = 4). Additionally, access to good breeding stock, lack of animal records, and an established marketing chain were also mentioned in all the three countries. Ectoparasites, gastrointestinal parasites, orf, and sheep/goat pox were the most frequently mentioned disease challenges causing productivity losses. Many articles provided potential solutions as suggested by farmers, including improved access to veterinary services and medicines, improved record keeping, and access to good breeding stock. Farmers highlighted the value of community-based participatory development plans to increase education on disease control, land management, and husbandry. This review also highlighted knowledge gaps, the need for further research, particularly in Tanzania and Uganda, and the importance of addressing multiple challenges holistically due to the links between constraints.

## 1. Introduction

Sustainable food production is an increasingly important challenge for the world’s expanding population. Population growth and increased consumer demand in developing countries, including East Africa, has resulted in an increase in the consumption of animal products such as meat and dairy [1,2]. Consequently, sustainable livestock production has an important role in food and environmental security [3,4,5,6,7].

Sheep and goats are an important commodity for smallholder farmers across East Africa and play an important role for home consumption, are a source of cash income for products such as meat, milk, wool, hides, and manure, and have a significance in their social value [8,9]. Additionally, small ruminants are of benefit to smallholder farmers because of their adaptation to harsh environments and their reproductive success with a short gestation period [10,11]. In 2017, in Ethiopia, there were estimated to be approximately 30.7 million sheep and 30.2 million goats [11]; in 2016, in Tanzania—5 million sheep and 16.7 million goats [12]; and in 2018, in Uganda—3.4 million sheep and 12.3 million goats [13]. Indeed, the number of smallholder farmers that rely on livestock for their livelihood continues to grow [3,4,5].

However, many constraints remain to small ruminant production, including limitations in access to animal health products and services [10,12,14], a lack of good quality grazing due to bush encroachment and urbanisation [15,16,17], increasing episodes of drought [18], ineffective disease control [19,20,21], and limited access to markets [22]. Additionally, funding for the livestock sector is often under-represented and under-appreciated. Consequently, many poor livestock owners remain trapped in poverty, without the interventions that may enable their development [23,24]. Much of the associated published literature is disease/country-specific, focuses only on general constraints to production, or provides assessment of individual control or development programs.

Therefore, this systematic review aimed to identify the challenges associated with small ruminant production in the East African countries covered by the African Livestock Productivity and Health Advancement (ALPHA) initiative: Ethiopia, Tanzania, and Uganda. The ALPHA Initiative was developed in 2017, co-funded by the Bill & Melinda Gates Foundation (BMGF) and Zoetis lnc., with the aim of improving livestock production in sub-Saharan Africa [14]. Additionally, the authors aimed to review previously reported interventions and provide a summary of realistic practical solutions. It is anticipated that the constraints and potential solutions identified in this review may be utilised to inform selection of appropriate and practical interventions that may result in a lasting improvement in the small ruminant production sector throughout East Africa.

## 2. Materials and Methods

Preferred Reporting Items for Systematic Reviews and Meta-Analyses (PRISMA) guidelines were used for this review [25]. Three scientific databases (Web of Science, PubMed, and Embase) were utilised for the search of research articles performed on the 9th October 2019 for the publication period 1990–2019. The literature search, extraction of data, and analysis were performed by one author with advice sought from co-authors when appropriate. The search terms used to identify research articles on all three databases were: ((small ruminant OR goat OR sheep) AND (Tanzania OR Ethiopia OR Uganda) AND (product * OR econom *)). Figure 1 shows the inclusion and exclusion criteria: the literature retrieval process included the removal of duplicate articles and the exclusion of records by first screening the title and abstracts and then full texts of articles. Studies included must (i) have been performed in either Ethiopia, Tanzania or Uganda; (ii) have been focused on small ruminants (sheep or goats); (iii) have provided original quantitative or qualitative information on constraints to production; and (iv) have been available in the English language.

Data extracted from eligible articles were recorded in an Excel spreadsheet (Appendix A) and included (i) type of study (e.g., original study); (ii) sample size; (iii) sources of data (e.g., farmer survey); (iv) factors/constraints identified; and (v) recommendations for improvement.

Additionally, for the ranking analysis, eligible articles were selected that provided farmer-reported constraints to small ruminant production in order of importance, collected by utilising household surveys, participatory rural appraisal (PRA), or focus group discussions. From each of these selected articles, the top three most important constraints were identified, and indices were calculated for each constraint according to the formula reported previously [26,27,28]:(1)$$I_{j}=\overset{3}{\underset{i=1}{\sum }}r_{j}X_{ij}/\left(\overset{n}{\underset{j=1}{\sum }}\overset{3}{\underset{i=1}{\sum }}r_{j}X_{ij}\right)$$

A score/weight (*r_i_*) of 3, 2, and 1 was assigned to the top three most important constraints as rank (*i*) 1, 2, and 3, respectively. *X_ij_* is the number of results from articles with the rank *i* (*i* = 1, 2 or 3) to constraint *j*, where *j* = disease, feed shortage, drought, poor marketing, access to water, access to grazing land, access to breeding stock, access to veterinary services, theft, small flock size, lack of record keeping, lack of knowledge/education, and predators. Consequently, the highest index gives the constraint with the greatest importance. A Spearman’s rank correlation test was performed using R 3.6.0 [29] within RStudio [30] to measure the association of each country’s ranking of constraints with the overall ranking.

## 3. Results

Data for this review were extracted from a total of 54 articles after screening 1053 articles for duplications and eligibility based on the inclusion and exclusion criteria (see Figure 1). Many of the excluded articles were not focused on the target countries or on the target species, were seroprevalence studies, or experimental evaluations of diagnostic tests and feed supplements.

Articles were published between 1993 and 2019 (Figure 2), and most articles focused on research performed in Ethiopia (n = 44, 84.5%), with only 6 studies performed in Tanzania and 4 in Uganda. The spatial distribution of the studies performed in each country by region or district is shown in Figure 3. A total of 21/54 (38.9%) studies focused on goats only, 6/54 (11.1%)—on sheep only, and 27/54 (50%)—on small ruminants in general. Most articles (94.4%) described original studies performed in the target countries, with two review articles [11,31] and one household modelling study [32]. Of the original studies, 82.4% utilised household surveys, participatory rural appraisal (PRA), or focus group discussions which provided information on the challenges and constraints of small ruminant production.

### 3.1. Constraints to Small Ruminant Production

The constraints to small ruminant production that were identified in the eligible articles are shown in Figure 4. Disease/parasite infestation was the most commonly identified constraint to small ruminant production, with over half of all the studies (61.1%) and those performed in Ethiopia (51.9%) mentioning disease as a major challenge. Additional constraints identified in all the three countries include lack of farmer knowledge/education, limited access to good breeding stock, lack of animal records, limited access to veterinary services, and lack of an established marketing chain. Drought, shortage of feed, availability of grazing land, and predation were also mentioned as challenges in Ethiopia and Uganda, but not in the studies from Tanzania. The most frequently identified constraints in Ethiopia were disease (n = 28) and shortage of feed (n = 13), in Tanzania—disease (n = 3) and lack of farmer knowledge (n = 3), and in Uganda—access to veterinary services (n = 4).

### 3.2. Farmer-Reported Ranking of Constraints

A total of 16/54 articles (11 from Ethiopia [9,10,15,21,33,34,35,36,37,38,39], 1 from Tanzania [40], and 4 from Uganda [19,28,41,42]) that reported constraints to small ruminant production in order of importance by farmers were selected for ranking analysis. Six articles included studies from multiple districts/regions which comprised different altitudes or production systems, and therefore for these articles each study was considered as a separate result (n = 25) [9,10,28,33,37,39]. Table 1 shows the rankings of each constraint for the three countries. The top 3 constraints were different for each of the three countries. In both Ethiopia and Uganda, which accounted for 24/25 of the studies, the biggest concern to farmers in relation to small ruminant production was the impact of disease (I = 0.292 and 0.441, respectively). There was only one study performed in Tanzania where the most important challenge was small flock size (I = 0.500) which was important due to the lack of genetic diversity available for breeding [40]. These results were observed to be significantly different from those of the overall result for all countries (Spearman’s rho statistic = −0.632, *p* = 0.021). The second and third constraints identified in Ethiopia were feed shortage and drought, in Tanzania—lack of record keeping and lack of knowledge/education, and in Uganda—access to grazing land and access to veterinary services, respectively.

### 3.3. Disease

The specific diseases/parasites identified as constraints to small ruminant production and the articles that mention each disease are shown in Table 2. Ectoparasites, including ticks, fleas, lice, and mites, were the most frequently mentioned challenge causing productivity losses in Ethiopia. Gastrointestinal parasites and their intermediate stages (including *Haemonchus* spp., *Trichostrongylus* spp., *Cysticercus tenuicollis*), and sheep/goat pox were the second and third most frequently mentioned diseases in Ethiopia, respectively. Two of the four studies performed in Uganda [19,28] identified 5 diseases (ectoparasties, gastrointestinal parasites, orf, contagious caprine pleuropneumonia [CCPP], and heartwater), and 3/6 studies performed in Tanzania [43,44,45] identified 3 diseases (brucellosis, peste des petits ruminants (PPR), and foot-and-mouth disease (FMD)) as constraints to small ruminant production.

Ectoparasites and sheep/goat pox were reported to cause symptoms such as itching, inflammation, anaemia, and poor body condition. They were reported to affect skin and hide quality resulting in skin rejection and therefore reduced income for farmers [22,37,46,47,49,50,52]. Additionally, diseases transmitted via ectoparasites such as heartwater and Nairobi sheep disease, were reported in Ethiopia and Uganda as causing loss of body condition, fever, gait and respiratory problems [57]. Gastrointestinal parasites were associated with poor body condition, diarrhoea, slow growth of lambs and were reported as the main cause of sheep deaths [38,55,56]. *Toxoplasma gondii* seropositivity was reported to be significantly associated with abortion [64] and condemnation of affected organs or carcasses was reported at the slaughterhouse due to the presence of hydatid cysts [54]. Viral infections such as sheep and goat pox, orf, PPR, and bacterial infections such as pasteurellosis, brucellosis, and CCPP were reported to cause symptoms such as abortion (brucellosis) [61,62], skin defects (sheep and goat pox and orf) [46,51], and a drop in milk production (FMD and mastitis) [45,63].

### 3.4. Potential Solutions to the Challenges Identified

A total of 40/54 of the eligible articles also provided potential solutions to the constraints identified above and are described in Table 3. Six of these articles performed an assessment of the impact of animal welfare or veterinary improvement projects, such as the Dairy Goat Development Programme (DGDP) implemented in Ethiopia in 1989 [65,66] and a project coordinated by Heifer International in Tanzania [31], which both aimed to promote goat keeping to smallholder farmers. Additionally [8], it provided an in-depth analysis of a dairy goat cooperative in Tanzania (Twawose) with the aim of assessing whether smallholder livelihoods were enhanced through the commercialisation of goat milk yoghurt. Two additional community-based projects that were assessed in Ethiopia included a community-based animal health worker (CAHW) scheme [67] and a community-based breeding programme (CBBP) [11], both of which utilised a participatory approach and considered farmers’ needs and views to improve livestock health and genetics, respectively.

## 4. Discussion

This review aimed to identify the challenges faced by smallholder farmers for small ruminant production in the three East African countries covered by the ALPHA initiative and to identify potential solutions that could be implemented in development projects in the future. Twenty-five constraints to small ruminant production were identified in the eligible articles, with disease or parasite infestation, shortage of feed and grazing land, and access to water or drought being commonly identified.

Thirty-three of the 54 reviewed papers specifically investigated general constraints to small ruminant production. Of the remaining 21 papers, 14 focused upon a specific disease, presenting results from seroprevalence studies and identifying putative risk factors. Three papers were focused on the impact and prevalence of ectoparasites and four papers considered multiple diseases. Disease was commonly identified as a major limitation to small ruminant production, which may reflect a reporting or publication bias and therefore the number of publications cannot be used to indicate importance. Indeed, neither disease nor any disease-related term was included within the search terms. Additionally, although these remaining 21 papers did give some consideration to the challenges of small ruminant production, our results may have been biased towards prioritisation of parasitism by these studies. Indeed, discrepancies may arise in the estimation of the importance of disease impacts when studies are based on different reporting measures, for example, between expert and farmer opinions [24]. For those studies that included a ranking of constraints, various approaches were utilised to collect the information and perform the analysis. The ranking analysis reported in this review can be considered as the perceived importance of constraints to small ruminant production amongst farmers included within the reviewed studies, although extrapolation to a wider population may not be justified.

The presence of ectoparasites and gastrointestinal parasites was frequently identified in small ruminants by many of the articles. Recommendations obtained in this review suggest that prevention and control strategies such as the use of dips and sprays or anthelmintics need to be affordable and sustainable and require consideration of seasonal parasite dynamicity [20] and the increasing likelihood of resistance due to misuse and/or overuse [56]. Many of the articles in this review utilised participatory approaches to disease surveillance, which are useful to understand disease prevalence, but also the impact of disease and the social implications involved when designing improvement programmes [70,71]. Although this method is useful in combination with molecular or serological studies, consideration is required as to the ability of farmers to correctly diagnose disease based on clinical signs alone. Consequently, further epidemiological studies are required on major livestock diseases in all the three countries. Emphasis should also be placed on effective disease surveillance, especially in the regions that border neighbouring countries, so that transboundary transmission of disease can be rapidly controlled. Additionally, isolation and characterisation of the infectious agent to determine the route of transmission is important, and the use of predictive models may help to inform control strategies including ensuring livestock are vaccinated with appropriate circulating strains [20,45].

Results from this review demonstrate a requirement for improved veterinary services and infrastructure, as at least one article from each of the three countries mentioned access to veterinary services as a constraint to small ruminant production. Without adequate local veterinary services, farmers often take alternative measures to improve the health of their animals, such as obtaining illegal or inappropriate drugs or visiting traditional healers [21]. Improved diagnostic capability, facilities, and medicine supply has been suggested in many of the articles. Additionally, public awareness and education programmes on basic animal husbandry, biosecurity, and control may help to minimise disease transmission at the farm level [54].

This review has highlighted the need for a multidisciplinary approach to improve small ruminant production. Previous studies performed in various African countries have suggested that solutions should not target specific constraints to livestock production in isolation [72,73]. Results from this review were in agreement. For example, a study by Mayberry et al. [32] demonstrated that although better healthcare improved the productivity of goats, the biggest improvements were seen when coupled with improved nutrition. Additionally, the Dairy Goat Development Programme (DGDP) implemented in Tanzania demonstrated that although the programme may have resulted in genetic improvement by improved recording and a reduction in in-breeding [58,65], these changes were only beneficial when animals were well-nourished and received basic healthcare.

Several community-based improvement schemes were shown to be effective in improving awareness and knowledge of farmers on animal health and consequently improving productivity. For example, the use of community-based animal health workers (CAHW) in Ethiopia [67] that are selected by the community to receive basic training on disease, vaccination, and the treatment of minor ailments, were considered highly trustworthy, accessible, and affordable for farmers. These workers are especially useful as they already possess the indigenous knowledge of disease presentation and social culture and can disseminate information to the community. This may be particularly useful for mobile pastoralists that may sometimes be excluded from surveillance studies and development programmes [74]. The formation of farmer associations/cooperatives can provide a participatory approach to development and may be useful for improving access to markets, for processing products such as milk, for the management of land, for the provision of credit, for the rotation of breeding bucks and maintaining records, and as opportunities for training, among others [11,31]. Indeed, the advantages of dairy cooperatives have been demonstrated in both East Africa and India [75,76,77,78]. Additionally, the use of community-based improvement schemes for land management may be a potential solution to encourage environmental rehabilitation, the conservation of natural resources, and access to feed supplements, with the overall goal of improving livestock nutrition [15,79]. A coordinated approach may be useful so that participants along various levels of the value chain might see a benefit, from sheep producers to agri-business entrepreneurs [80].

Most (84.5%) of the articles eligible for this review involved studies performed in Ethiopia, with studies performed in all but one administrative region. Knowledge gaps exist in Tanzania and Uganda, where studies have been performed in only a small number of regions/districts, and therefore there may have been some study bias, for example, with respect to the study areas. Consequently, further studies may be required focusing on specific locations. Additionally, it is possible that as only those articles were selected that were available in the English language, this may have excluded valuable data. Most of the eligible articles included original studies directly targeting small ruminant farmers utilising household surveys or participatory approaches or analyses of implemented development programmes. Consequently, the potential solutions identified are those that have been tried and tested on a small scale or have been directly suggested by farmers as improvements that would enhance their livelihoods. Results from the three countries targeted for this review indicate that similar challenges to small ruminant production exist, and therefore the solutions identified could be implemented throughout East Africa where required. Indeed, a study interviewing pastoralists in Northern Kenya also identified many of the same challenges to sheep and goat production (i.e., disease, drought, predators, lack of veterinary services) [81]. However, it is important that solutions are tailored due to the variety of farming practices and cultural traditions. For example, although Nigeria is one of the four countries covered by the ALPHA initiative, it was not included in this review due to the geographical distance and therefore potential differences in factors such as culture, farming systems, climate, among others.

## 5. Conclusions

In conclusion, this review sought to identify constraints to small ruminant production in the East African countries covered by the ALPHA initiative and has identified gaps in knowledge, particularly in Tanzania and Uganda, highlighting the requirement for further research in these areas. This review has highlighted the importance of providing sustainable solutions with input from farmers via the use of participatory approaches and has included some of these recommendations that could be implemented in the target countries. However, it is important that in the creation of development programmes, constraints should not be targeted in isolation due to the links between them. For example, there may be little benefit from improving the performance of a goat by genetics alone if access to basic healthcare, disease prevention, and nutrition is not readily available. Consequently, future development programmes should aim to address multiple challenges holistically so that increased production may enhance the livelihoods of small ruminant farmers. Additionally, a novel methodology for the measurement of impact should be considered due to the driver of many livestock producers being subsistence over cash income.

## Figures and Tables

**Figure 1 vetsci-08-00005-f001:**
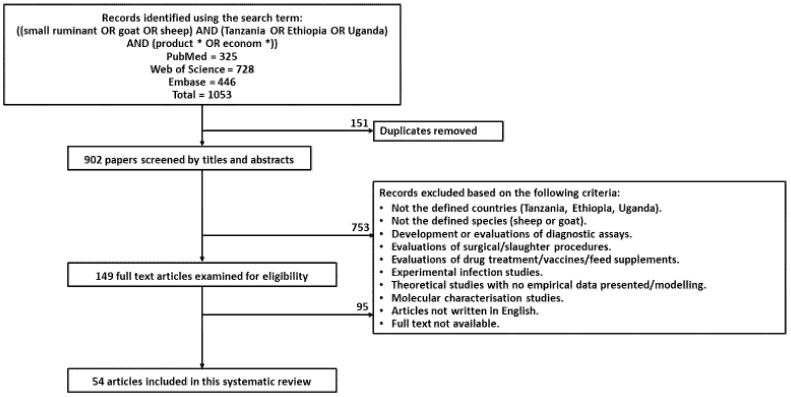
Flowchart for the systematic review to identify eligible articles to determine the constraints influencing small ruminant production.

**Figure 2 vetsci-08-00005-f002:**
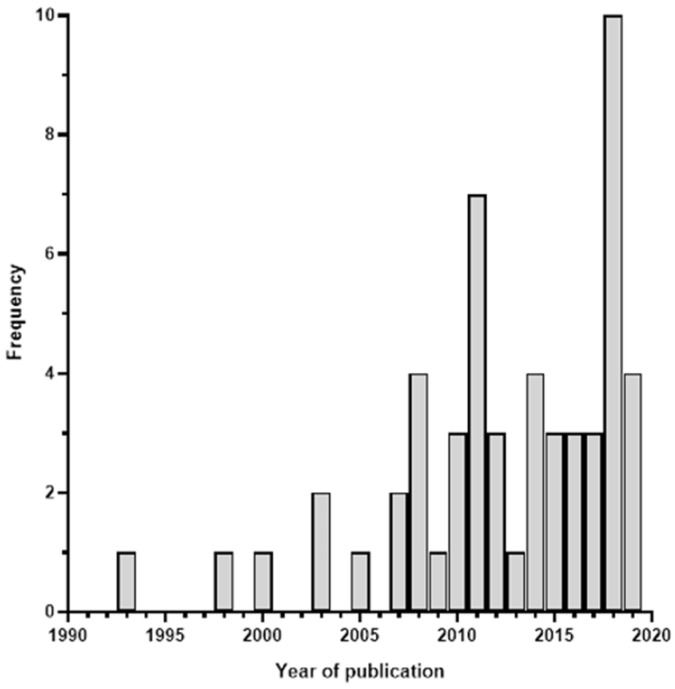
Number of articles and the year of publication.

**Figure 3 vetsci-08-00005-f003:**
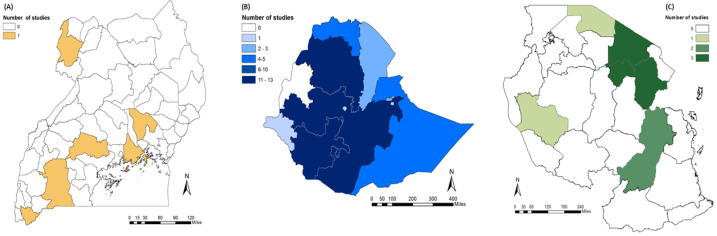
The distribution of studies performed in the 54 reviewed articles. Map (**A**) represents the administrative districts of Uganda. Maps (**B**,**C**) represent the administrative regions of Ethiopia and Tanzania, respectively. Some articles performed studies in more than one region/district.

**Figure 4 vetsci-08-00005-f004:**
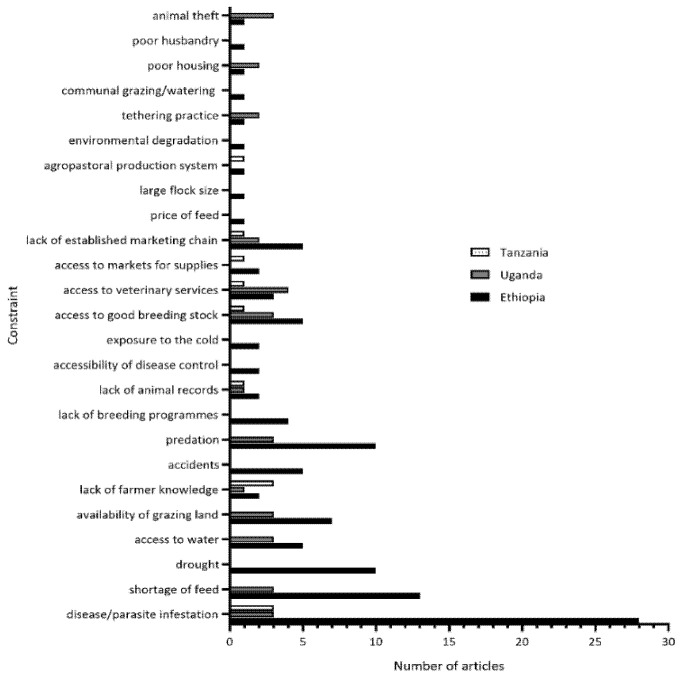
Constraints to small ruminant production identified from eligible articles in Ethiopia, Tanzania, and Uganda.

**Table 1 vetsci-08-00005-t001:** Farmer-reported ranking of constraints to small ruminant production.

Constraint	Ethiopia				Tanzania				Uganda				Overall			
	a = 11, n = 18				a = 1, n = 1				a = 4, n = 6				a = 16, n = 25			
	I1	I2	I3	Index	I1	I2	I3	Index	I1	I2	I3	Index	I1	I2	I3	Index
Disease	7	3	4	0.292	0	0	0	0.000	5	0	0	0.441	12	3	4	0.311
Feed shortage	6	4	3	0.274	0	0	0	0.000	0	0	0	0.000	6	4	3	0.196
Drought	1	5	3	0.151	0	0	0	0.000	0	0	0	0.000	1	5	3	0.108
Poor marketing	1	1	1	0.057	0	0	0	0.000	0	0	0	0.000	1	1	1	0.041
Access to water	1	3	3	0.113	0	0	0	0.000	0	0	1	0.029	1	3	4	0.088
Access to grazing land	2	0	0	0.057	0	0	0	0.000	1	4	0	0.324	3	4	0	0.115
Access to breeding stock	0	1	1	0.028	0	0	0	0.000	0	0	1	0.029	0	1	2	0.027
Access to veterinary services	0	0	0	0.000	0	0	0	0.000	0	1	2	0.118	0	1	2	0.027
Theft	0	0	0	0.000	0	0	0	0.000	0	0	2	0.059	0	0	2	0.014
Small flock size	0	0	0	0.000	1	0	0	0.500	0	0	0	0.000	1	0	0	0.020
Lack of record keeping	0	0	0	0.000	0	1	0	0.333	0	0	0	0.000	0	1	0	0.014
Lack of knowledge/education	0	0	0	0.000	0	0	1	0.167	0	0	0	0.000	0	0	1	0.007
Predators	0	1	1	0.028	0	0	0	0.000	0	1	0	0.059	0	2	1	0.034
Spearman’s rho statistic	0.939 (*p* = < 0.001)				−0.632 (*p* = 0.021)				0.318 (*p* = 0.289)							

a = number of articles. n = number of ranking results (some articles include multiple results). I1 = total number of studies identifying this constraint as the most important by farmers. I2 = total number of studies identifying this constraint as the second most important by farmers. I3 = total number of studies identifying this constraint as the third most important by farmers. Underlined Index values represent the three highest ranking constraints.

**Table 2 vetsci-08-00005-t002:** Number of articles mentioning each disease as a constraint to small ruminant production.

Disease	Ethiopia	Uganda	Tanzania	Total (% ^1^)	References
Ectoparasites	12	1	0	13 (24.1%)	[10,19,21,22,33,37,46,47,48,49,50,51,52]
Gastrointestinal parasites	9	2	0	11 (20.4%)	[19,28,33,34,37,38,48,53,54,55,56]
Orf	7	1	0	8 (14.8%)	[9,19,37,48,51,56,57,58]
Sheep/goat pox	7	0	0	7 (13.0%)	[9,10,20,21,33,46,58]
Pasteurellosis	6	0	0	6 (11.1%)	[9,10,20,33,37,48]
CCPP	5	1	0	6 (11.1%)	[9,10,19,20,33,50]
Brucellosis	5	0	1	6 (11.1%)	[20,43,59,60,61,62]
PPR	4	0	1	5 (9.3%)	[9,10,20,44,58]
Coenurosis	4	0	0	4 (7.4%)	[37,48,56,58]
Diarrhoeal syndrome	4	0	0	4 (7.4%)	[10,34,50,56]
Anthrax	3	0	0	3 (5.6%)	[10,37,50]
Pneumonia	3	0	0	3 (5.6%)	[34,37,57]
Mastitis	3	0	0	3 (5.6%)	[10,21,63]
Heartwater	1	1	0	2 (3.7%)	[19,57]
Mineral deficiency	2	0	0	2 (3.7%)	[10,38]
Listeriosis/circling disease	1	0	0	1 (1.9%)	[10]
FMD	0	0	1	1 (1.9%)	[45]
Foot rot	1	0	0	1 (1.9%)	[21]
Nairobi sheep disease	1	0	0	1 (1.9%)	[20]
Pyogenic infection	1	0	0	1 (1.9%)	[57]
*Toxoplasma gondii*	1	0	0	1 (1.9%)	[64]
Trypanosomiasis	1	0	0	1 (1.9%)	[21]
Caseous lymphadenitis	1	0	0	1 (1.9%)	[57]

^1^ n = 54. CCPP: contagious caprine pleuropneumonia; PPR: peste des petits ruminants; FMD: foot-and-mouth disease.

**Table 3 vetsci-08-00005-t003:** Potential recommendations to improve small ruminant production in Ethiopia, Uganda, and Tanzania.

Constraint	Solution	References
Disease	Effective control of ectoparasites using dips or sprays	[19,21,33,37,46,52]
	Sustainable control options for gastrointestinal parasites, for example, by using medicinal plants, and controlled use of anthelmintics to reduce the likelihood of resistance	[54,56]
	Improve education for farmers and animal health workers on disease transmission and biosecurity	[31,43,47,54,58,61]
	Regular epidemiological surveys of transboundary animal diseases should be performed with emphasis on the borders with neighbouring countries	[20]
	Investment in the development of effective vaccines and the implementation of vaccination programmes, e.g., for brucellosis, FMD, sheep and goat pox, and heartwater	[21,45,51,57,60]
Access to veterinary services	Establish more localised veterinary centres for improved access to routine and preventative veterinary care	[10,32,41,44,58]
	Increase training for veterinary professionals including paravets	[9]
	Improve the medicine supply system and control	[9]
	Use of community-based animal health workers (CAHW) or participatory groups that are provided with in-depth training in animal health and husbandry, e.g., to perform vaccinations and treat minor ailments	[8,9,31,65,67]
Availability of quality feed	Education on managing and preserving pastures	[41]
	Increase grazing land availability through environmental rehabilitation and conservation of natural resources	[15]
	Improve forage quality through controlled grazing, reseeding, the introduction of adaptable forage species and improved fodder grasses and legumes	[33,38,41]
	Supplementation of small ruminant diets with leguminous tree foliage with a high crude protein content or other feed supplements such as noug seed cake	[32,34,42]
Drought	Early warning systems for drought using predictive models	[19]
	Movement of livestock in drought situations to minimise livestock pressure on natural resources	[35]
Breeding programmes	Rotation of breeding males/mixing of flocks to reduce inbreeding	[58]
	Community-based sustainable breeding improvement programmes with a focus on indigenous/hybrid stock and with input from farmers’ experience and their trait preferences	[10,11,37,41,58,65,66,68]
	Availability of cheap, easily accessible, and simple-to-use reproductive technologies, including access to artificial insemination	[11]
	Improved record keeping of breeding performance with input from all family members (see below)	[58]
	Training on financial and technical management of breeding programs with support from national research institutions	[11]
Record keeping	Training of farmers on the importance of animal identification	[40]
	Improved supply of affordable ID materials	[40]
	Improved easy-to-use record keeping databases, e.g., for use on a mobile phone	[11]
Availability of markets	Perform value chain analysis to understand marketing barriers such as access and smallholder participation	[10,22]
	Improvement of infrastructure and information distribution	[22]
	Incentives for smallholder producers to invest in improving output of desirable animal products	[65]
	Establishment of farmer groups, associations, or cooperatives (see above) to increase links and participation in formal markets and increased access to information	[10,11,22]
Predation	Use of shepherding should be promoted among smallholder farmers to reduce predation by wild animals	[69]

## Data Availability

Data is contained within the article or Appendix A.

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
