# Peer review of "Small Ruminant Production in Tanzania, Uganda, and Ethiopia: A Systematic Review of Constraints and Potential Solutions"

_vetsci, 2020, doi:10.3390/vetsci8010005_

Round 1

Reviewer 1 Report

Satisfactory corrections
Best regard

Reviewer 2 Report

Authors must accept changes to the manuscript before submitting it. My recommendation for this manuscript is accepted in present form.

This manuscript is a resubmission of an earlier submission. The following is a list of the peer review reports and author responses from that submission.

Round 1

Reviewer 1 Report

I have read the manuscript with much interest. It is a nice study with a clear description of the methods used and a critical description of the results found. I have only few remarks that should be addressed:

Lines 147-149

“Gastrointestinal parasites (including Haemonchus spp., Trichostrongylus spp., Taenia spp., Echinococcus granulosus) and sheep/goat pox were the second and third most frequently mentioned diseases in Ethiopia, respectively.”

  • Gastrointestinal cestodes in ruminants don’t belong to Taenia spp, but are Moniezia spp or Stilesia spp, as intestinal taeniasis can only be acquired by carnivorism. Ruminants can be intermediate hosts of Taenia spp at abdominal (T hydatigena), muscle (T ovis) or other tissue locations (T serialis).
  • Ruminants are intermediate hosts of Echinococcus granulosus and thus don’t develop an intestinal infection but hydatid cysts in different tissues, mainly in the liver and lungs.

Table 2: correct “trypanasomiasis” to “trypanosomiasis”

Lines 164-166: “Toxoplasma gondii seropositivity was reported to be significantly associated with abortion and condemnation of 165 affected organs or carcasses was reported at the slaughterhouse due to the presence of cysts [52].”

  • This should be verified. Toxoplasma tissue cysts are max 100µm of diameter and thus not visible during meat inspection. Reference 52 is on Cestode infections, not on Toxoplasma

Reviewer 2 Report

A review is a good starting point for identifying critical issues and on the basis of which to implement synergistic development programs. The article presents some points for improvement before being published.
- the A.L.P.H.A. it includes 4 countries of Sub-Sarian Africa but in the work we only refer to 3, it would be appropriate to include Nigeria as well or in any case illustrate the valid reasons that led to its exclusion.
- the introduction must be improved in order to better frame the problem addressed.
- in "Materials and Methods" (lines 59-60) it is wrong to report the tasks of each author, there is a specific section in the manuscript (Authors' contributions).
- reorganize the "Discussions" section, it would be advisable to report results and discussions in a single section to avoid reporting entire sentences from the "Results" section.
- improved discussions by expanding the bibliography, also with research and projects carried out in other African countries, to suggest a greater number of potential solutions
- standardize the bibliography section to the journal format